# PTPRD and DCC Are Novel BACE1 Substrates Differentially Expressed in Alzheimer’s Disease: A Data Mining and Bioinformatics Study

**DOI:** 10.3390/ijms23094568

**Published:** 2022-04-20

**Authors:** Hannah A. Taylor, Katie J. Simmons, Eva M. Clavane, Christopher J. Trevelyan, Jane M. Brown, Lena Przemyłska, Nicole T. Watt, Laura C. Matthews, Paul J. Meakin

**Affiliations:** 1Leeds Institute of Cardiovascular and Metabolic Medicine, Faculty of Medicine and Health, University of Leeds, Leeds LS2 9JT, UK; um19hat@leeds.ac.uk (H.A.T.); k.j.simmons@leeds.ac.uk (K.J.S.); umemc@leeds.ac.uk (E.M.C.); bs16cjt@leeds.ac.uk (C.J.T.); j.m.y.brown@leeds.ac.uk (J.M.B.); ed18l4p@leeds.ac.uk (L.P.); n.t.watt@leeds.ac.uk (N.T.W.); 2Leeds Institute of Medical Research, Faculty of Medicine and Health, University of Leeds, Leeds LS2 9JT, UK; l.c.matthews@leeds.ac.uk

**Keywords:** Alzheimer’s disease, beta secretase 1 (BACE1), Netrin receptor DCC, protein tyrosine phosphatase receptor type D (PTPRD), BACE1 substrates

## Abstract

The β-site Amyloid precursor protein Cleaving Enzyme 1 (BACE1) is an extensively studied therapeutic target for Alzheimer’s disease (AD), owing to its role in the production of neurotoxic amyloid beta (Aβ) peptides. However, despite numerous BACE1 inhibitors entering clinical trials, none have successfully improved AD pathogenesis, despite effectively lowering Aβ concentrations. This can, in part, be attributed to an incomplete understanding of BACE1, including its physiological functions and substrate specificity. We propose that BACE1 has additional important physiological functions, mediated through substrates still to be identified. Thus, to address this, we computationally analysed a list of 533 BACE1 dependent proteins, identified from the literature, for potential BACE1 substrates, and compared them against proteins differentially expressed in AD. We identified 15 novel BACE1 substrates that were specifically altered in AD. To confirm our analysis, we validated Protein tyrosine phosphatase receptor type D (PTPRD) and Netrin receptor DCC (DCC) using Western blotting. These findings shed light on the BACE1 inhibitor failings and could enable the design of substrate-specific inhibitors to target alternative BACE1 substrates. Furthermore, it gives us a greater understanding of the roles of BACE1 and its dysfunction in AD.

## 1. Introduction

### 1.1. The Role of BACE1 in Alzheimer’s Disease 

β-site Amyloid precursor protein Cleaving Enzyme 1 (BACE1) is an aspartyl protease enzyme known for its role in generation of amyloid beta (Aβ) peptides. The accumulation of neurotoxic Aβ into amyloid plaques in the brain is believed to be a major driver of AD, through disruption of neuronal and synaptic function [1,2]. The role of BACE1 in Aβ generation via cleavage of the amyloid precursor protein (APP) is well established and widely researched. The discovery of genetic mutations in APP promoting AD pathology (Swedish mutation KM/NL and an Italian variant A673V) and the protective Icelandic (A673T) mutation provide compelling evidence that reducing Aβ production would be beneficial for preventing AD development [3,4,5]. However, current therapeutic treatments to minimise plaque levels, via increasing plaque removal with Aducanumab, an amyloid plaque targeting monoclonal antibody, do not appear to be effective [6,7]. Furthermore, despite extensive research, all clinical trials investigating BACE1 inhibitor treatment for AD have been discontinued, often failing owing to futility or safety issues [8]. This can be partly attributed to the lack of understanding surrounding additional functions of BACE1. For example adverse, non-amyloid dependent, phenotypes have been observed in both clinical trials and BACE1 null mice, including impaired axon organisation [9], synaptic [10], cognitive functions [11], and weight loss [12]. Therefore, the unknown alternative functions of BACE1 may contribute to the failures of BACE1 inhibitors, and a greater understanding of the physiological roles may reveal the reasons behind their failure and highlight new avenues to investigate.

### 1.2. Neuronal Roles and Substrates of BACE1

It is arguable that a greater understanding of the functional roles and substrates of BACE1 is the key to successful therapeutic targeting for treatment of AD. It is evident from inhibition, downregulation, and knockout studies, that BACE1 plays an important role in the brain. BACE1 activity is associated with the regulation of myelination, learning and memory, and synaptic function [1]. The lack of success with BACE1 inhibitors has been in part attributed to the role that BACE1 plays in synaptic vesicle docking, and therefore communication between neurons [13]. 

Despite being known for its cleavage of the APP protein, it is well acknowledged that BACE1 has low substrate specificity for APP and has numerous other known substrates. The roles of BACE1 outside of Aβ-production may contribute to the lack of success of therapeutic inhibition. Numerous secretome studies have investigated novel BACE1 substrates, and nearly 70 BACE1 substrates have been identified to date [14,15,16]. Currently, only a small number of the identified BACE1 substrates have been fully characterised, which have important functions in the brain, reviewed elsewhere [17,18]. The roles BACE1 plays in brain function, therefore, pose a limitation in therapeutically targeting it across the blood–brain barrier (BBB), and supports the argument against generic BACE1 inhibitors in order to avoid undesirable BACE1-mediated effects. Substrate-specific BACE1 inhibitors prove a potential therapeutic treatment for AD; however, this is only possible if the key substrates of BACE1 are known. 

### 1.3. Non-Neuronal Roles and Substrates of BACE1

As BACE1 has been primarily studied in neurons, there may be important non-neuronal functions which have not yet been identified. Evidence of this is observed in the consequential phenotype of BACE1 knockout mouse models [19]. More recently, investigations into BACE1 have alluded to roles outside of neurons and the central nervous system, where important functions in metabolism and the cardiovascular system have highlighted how little is known about the physiological roles of BACE1. This includes cardiovascular and immune functions, glucose homeostasis, metabolic functions, and insulin signalling [20,21,22,23,24,25]. AD has long been associated with vascular function, and the contribution of vascular impairments to the development of AD is well established. However, more recently, it has been proposed that BACE1 may also contribute to central [26] and peripheral vascular function [20]. BACE1 is expressed in endothelial cells including in the BBB [27]. In AD, an increase in BACE1 activity can also lead to accumulation of Aβ peptide deposition in cerebral vessels, called cerebral amyloid angiopathy (CAA) [28]. CAA has been linked to BBB impairments, vascular dysfunction, reduced cerebrovascular blood flow, and capillary degradation [24,29,30], leading to accelerated cognitive decline [23]. The resulting reduction in vascular clearance exacerbates the accumulation of Aβ in blood vessels and surrounding neurons. Increased activity of BACE1 in endothelial cells of the brain vasculature may, therefore, play an important, and not fully characterised, role in neurovascular function in AD. Therefore, although not fully understood, it is clear that BACE1 is associated with vascular function. 

The metabolic effects of BACE1 could present a potential link between the increased risk of AD in individuals with metabolic disease. BACE1 has been linked to insulin signalling [21] and leptin signalling [31,32], and has been implicated in the development of diabetes [22]. It has also been linked to metabolic syndrome and obesity, with BACE1 inhibition being linked to reductions in body weight in both experimental [19] and clinical data [12]. 

BACE1 has also been linked to white blood cell recruitment to infection via cleavage of P-selectin glycoprotein ligand-1 (PSGL-1) [16], and abnormal inflammation via cleavage of interleukin-1 receptor II (IL-1R2) [25].

The focus of this study was, therefore, to investigate the potential roles of BACE1 outside of the production of Aβ peptides, and outside neurons, through identification of novel and uncharacterised BACE1 substrates and dependent proteins. In addition to numerous pathways and functions, we identified 15 AD-specific potential BACE1 substrates using multiple transcriptomic datasets. Increased BACE1 levels and activity are observed in AD, as well as other diseases associated with an increased risk of AD, such as diabetes; however, the mechanisms are not entirely clear. Given the more recently described roles of BACE1 in metabolism and cardiovascular disease, and the lack of knowledge surrounding BACE1 substrates, this computational analysis could present a first step into deciphering the roles of BACE1 in health, disease, and importantly in AD.

## 2. Results

### 2.1. BACE1-Regulated Proteins

To obtain a list of BACE1-regulated proteins, proteomics datasets comparing models with altered BACE1 expression/activity were collated, including BACE1 knockout, gain of function, loss of function, or chemical inhibition with the BACE1 inhibitor C3 (Figure 1) [14,33,34,35,36]. The BACE1-regulated protein list obtained was then compared by species and cell/tissue type to determine if there was a consensus list to take forward for further analysis. The list of ‘BACE1-regulated proteins’ were analysed for function using the Enrichr database, with the most common and significant terms including axon guidance, cell adhesion, cadherin signalling, lipoproteins, and synapse function (Figure 2).

### 2.2. BACE1 Substrates

To obtain a list of which proteins were directly regulated by BACE1, the list of 533 BACE1-regulated proteins was compared against a list of 934 bioinformatically predicted BACE1 substrates (Figure 3a) [37]. The overlap between these two groups gave a list of 120 proteins, found to be differentially expressed in response to changes in BACE1, and predicted to be BACE1 substrates (Table 1). Importantly, our analysis identified previously validated BACE1 substrates. This list was taken forward for functional analysis using the Enrichr database. The list of potential BACE1 substrates was again associated with: axon guidance, cell adhesion, synapse organisation, and also identified involvement with transmembrane tyrosine kinase and phosphatase activity (Figure 3b–d).

### 2.3. BACE1 Substrates Altered in Alzheimer’s Disease

To identify BACE1 substrates with potentially important but as yet unidentified roles in AD, the list of 120 BACE1 substrates was compared against a list of proteins differentially expressed in the brains of AD patients post-mortem, identified using single nucleus transcriptomics (Figure 4a) [38]. Interestingly this identified a list of 25 BACE1 substrates with altered expression in the brain in AD, in comparison to healthy brain tissue (Table 2). These 25 proteins were associated with axon guidance, cell adhesion, Slit/Robo signalling, Netrin-1 signalling, Aβ binding, and phosphatase activity (Figure 4b–d). 

To confirm which of these 25 proteins were changed specifically in AD, the list was compared against four other cerebral diseases (Figure 5a). This identified 15 proteins that are likely to be altered specifically in AD. Functional analysis of these 15 AD-specific BACE1 substrates found associations with Jak/STAT signalling, cell adhesion, and type 1 interferon pathway (Figure 5b,c). When input into STRING to investigate protein interactions, neuron generation and cell adhesion were the most prevalent functional associations (Figure 5d).

### 2.4. In Silico Modelling of Predicted Substrate Interactions

Our bioinformatic approach identified both proteins as potential in vivo substrates for BACE1 [37]. The region predicted to be the putative cleavage recognition sequence for PTPRD was KQLQSGQ (residues 1216–1223) and PVDTNLID (residues 1064–1071) for DCC which are both located in the extracellular domain. We used the High Ambiguity Driven protein–protein DOCKing (HADDOCK) protein–protein docking webserver [39,40] to predict the molecular basis for the interaction. 

For the BACE1-PTPRD interaction, cluster 1 (Figure 6a,b) contained 29 of 132 structures and the best HADDOCK score (−94.3 +/− 6.5). For the BACE1-DCC interactions, cluster 1 (Figure 6c,d) contained the majority of structures (109/154) and the best HADDOCK score (−101.3 +/− 1.8). These scores are similar in value as the best HADDOCK score for the predicted interaction of APP with BACE1 (179 structures in 8 clusters, which represents 89% of the water-refined models generated with the best score of −100.8 +/− 6.9). Taking these models and bioinformatics data together, we can have confidence that both PTPRD and DCC can interact with the active site of BACE1, supporting our findings that PTPRD and DCC are BACE1 substrates. 

### 2.5. Experimental Validation of Novel Substrates

To experimentally validate our analysis, one validated BACE1 substrate, and two novel BACE1 substrates were investigated using Western blotting. The analysis was conducted using murine primary brain microvascular endothelial cells (BECs) and a brain endothelial cell line (hCMEC/D3) as our focus was on identifying alternative BACE1 substrates and possible functions of BACE1 outside of neuronal cells. 

Using an N-terminus antibody we observed a significant increase (1.95-fold, *p* < 0.05) in cellular NCAM1 expression in BACE1^−/−^ BECs, when compared with WT cells (Figure 7A). A similar increase (1.48-fold, *p* < 0.05) was observed in hCMEC/D3 cells treated with a BACE1 inhibitor (Figure 7D), suggestive of reduced BACE1-mediated cleavage. Conversely, overexpression of BACE1 reduced full length NCAM1 (0.74-fold, *p* < 0.05) (Figure 7G). Taken together, this supports recent literature identifying that NCAM1 is cleaved by BACE1 [41]. 

We identified PTPRD as a novel BACE1 substrate, observing a significant 1.95-fold increase (*p* < 0.05) in PTPRD expression in BACE1^−/−^ BECs when compared with WT cells (Figure 7B) and a 1.54-fold increase in M-3 treated hCMEC/D3 cells (Figure 7E). Overexpression of BACE1 resulted in a 0.59-fold change (*p* < 0.05) in full length PTPRD (Figure 7H). This data demonstrates that PTPRD is cleaved by BACE1. 

DCC, also identified as a novel BACE1 substrate, was significantly increased 1.53-fold (*p* < 0.05) in BACE1^−/−^ BECs and following M-3 treatment (1.29, *p* < 0.05) (Figure 7C,F). BACE1 overexpression produced modest but non-significant changes in DCC expression in hCMEC/D3 (Figure 7I). PTPRD and DCC gene expression in hCMEC/D3 was not altered by BACE1 inhibitor treatment or overexpression (Figure 7J,K).

## 3. Discussion

BACE1 is an important therapeutic target for Aβ production in AD; however, its non-amyloid functions remain less well understood. This study compiled data from a range of sources to systematically identify novel protein substrates and target pathways controlled by BACE1 to address this lack of knowledge.

### 3.1. Novel BACE1-Regulated Proteins and Potential Substrates

A list of 533 BACE1-regulated proteins were identified, of which 120 were potential BACE1 substrates including numerous novel targets. As multiple sources were used to identify BACE1-regulated proteins, a low stringency was set which enabled a broad focus. As BACE1 is not a highly substrate-specific protease, the list of bioinformatically predicted substrates enabled greater confidence in the list being true BACE1 substrates [37,42]. This approach identified known substrates, including APP, CHL-1, L1CAM, SEZ6, and SEZ6L, as expected. It also identified NCAM1, a recently validated BACE1 substrate which we confirmed as having increased expression in our primary BACE1^−/−^ BECs via Western blot [41] (Figure 7A). The few BACE1 substrates that have been fully characterised have important functions in the brain, including axon guidance, myelination, and neurite outgrowth [17]. 

Previously identified BACE1 substrates have been shown to regulate functions including axon guidance, synaptic plasticity, and neurogenesis; our list of 120 BACE1 substrates has expanded this knowledge to include tyrosine kinase and phosphatase signalling and cell adhesion. The functional ontology analysis of BACE1-regulated proteins and substrates using Enrichr, showed associations with various pathways and functions. Functional analysis of the BACE1 substrates showed associations with transmembrane protein tyrosine kinase, axon guidance, and cell adhesion, all of which have been previously reported to be associated with BACE1 activity [43,44,45]. The identification of these BACE1-regulated proteins and substrates provided a strong platform to further explore additional, non-amyloid-dependent, effects of BACE1 in AD.

Using single cell and nucleus RNA sequencing data, we identified 15 BACE1 substrates which were specifically altered in human AD brain samples, when compared against other neurological diseases; stroke, TBI, EAE, and epilepsy [46]. A limitation of the comparison against other neurological diseases was the data for TBI, EAE, stroke, and epilepsy being endothelial-cell-specific. This was chosen as our experimental analysis used primary endothelial cells to validate BACE1 substrates. However, our list of proteins DE in AD included proteins from neuronal cells, as well as endothelial cells. It is therefore possible that some proteins were not identified in the other diseases due to their localisation in the brain. 

Although BACE1 is reported to have a relatively open active site and loose substrate specificity, using predicted cut sites of known substrates, Johnson et al. bioinformatically developed a scoring matrix to identify a list of potential BACE1 substrates [37]. The bioinformatic analysis of Johnson et al. predicted BACE1 cleavage recognition sequences (NKQLQSGQ) in PTPRD and (PVDTNLID) in DCC. While scores of 1.72 × 10^−5^ and 1.02 × 10^−5^ for PTPRD and DCC, respectively, are lower than APP (1 × 10^−3^) they are similar to validated BACE1 substrates, such as NRG1 and PSGL-1 [16,47]. Using these predicted BACE1 cleavage recognition sequences, we used in silico modelling to show that it would be possible for these proteins to interact with the active site of BACE1. This gives us confidence that PTPRD and DCC are likely to be true BACE1 substrates, rather than the observed changes being a result of alternative mechanisms. We validated PTPRD and DCC as novel AD-specific BACE1 substrates, via Western blotting, confirming our analysis. Further studies are required to confirm the additional BACE1 substrates and understand how BACE1 cleavage alters protein function.

### 3.2. Pathways and Proteins Associated with BACE1 in AD

#### 3.2.1. Receptor-Type Protein Tyrosine Phosphatases (PTPR)

This study identified seven members of the PTPR family as novel BACE1 substrates, with PTPRD specifically altered in AD. Protein tyrosine phosphatases are involved in a multitude of signalling pathways, often providing important regulation of signal transduction. In multiple stages of our analysis, protein tyrosine phosphatase and kinase terms were highly significant (Figure 3b and Figure 4d). Dysregulation of receptor-type protein tyrosine phosphatase signalling is frequently observed in disease, including diabetes and cardiovascular disease, which are associated with an increased risk of AD. The involvement of BACE1-mediated cleavage of PTPRD presents a potentially important novel mechanism underlying cardiometabolic disease and AD risk.

Although the literature on a relationship between BACE1 and PTPRD is minimal, it is more widely understood that BACE1 mediates the action of protein tyrosine phosphatase 1B (PTP1B). PTP1B is well established in insulin signalling and leptin signalling and has been implicated in the dysfunction of these signalling pathways observed in metabolic disease [31,48]. It, therefore, follows that a similar mechanism could occur between BACE1 and PTPRD. PTPRD localisation is reported to be largely restricted to the brain [49], and has been implicated as having roles in neuronal cell adhesion, synapse function, neurofibrillary pathology in AD, neurodevelopmental disorders, and cognitive impairments [50]. Although PTPRD has been associated with neurofibrillary tangles, the mechanism for this is not yet fully understood [51]. PTPRD is involved in pre- and post-synaptic differentiation of neurons, and synapse formation. PTPRD dephosphorylates TrkB and PDGFRβ, downregulating the MEK-ERK signalling pathway in neural precursor cells [52]. 

PTPRD is also associated with signal transducers and activators of transcription (STAT) signalling, including IL-6/JAK/STAT3 signalling. The IL-6/JAK/STAT3 signalling pathway is implicated in various diseases including cancer and diabetes [53]. Increased IL-6 induces phosphorylation of STAT3, resulting in signalling. However, IL-6-induced phosphorylation of STAT3 has also been shown to transiently upregulate PTPRD [54]. PTPRD then dephosphorylates STAT3, in turn negatively regulates IL-6/STAT3 signalling [55]. This is supported by the depletion of PTPRD by siRNA resulting in elevated STAT3 phosphorylation, and therefore signalling [56]. Our analysis shows for the first time that BACE1 cleaves the PTPRD pro-protein; therefore likely activating it. We propose that this activation of PTPRD by increased BACE1 activity results in PTPRD-mediated dephosphorylation of STAT3, leading to negative regulation of IL-6/STAT3 signalling. Significantly, memory impairments caused by Aβ have been linked to the inactivation of the JAK/STAT3 signalling pathway, with pSTAT3 levels age-dependently decreasing in AD [57]. Negative regulation of STAT3 signalling by BACE1, via PTPRD, concurs with the decrease in pSTAT3 observed in AD. This ultimately presents PTPRD as a novel and important substrate for BACE1 function, and suggests that specific targeting of BACE1-regulated signalling pathways may provide new avenues for therapeutic targeting of AD. It may also present a mechanism by which the increased risk observed between cardiovascular and metabolic disease, and dementia, may occur. This, therefore, questions what effect protein tyrosine phosphatases may have in AD, and whether associated changes in BACE1 activity may be having important and unknown impacts on brain function via the associated signalling pathways. 

#### 3.2.2. Netrin Receptor DCC

Netrin receptor DCC (DCC) was identified as a novel BACE1 substrate decreased in AD in our analysis. DCC is the receptor for Netrin-1, which together play an important role in the development of the nervous system through their role in the chemoattraction of axons and neurons [58]. In addition to the role in axon guidance and neurite outgrowth, DCC is proapoptotic and prevents tumour growth when it is not activated by Netrin-1. Conversely, when bound to Netrin-1, DCC acts as an oncoprotein, promoting cell survival. In a similar manner to its role in axon guidance, the Netrin-1/DCC interaction is also involved in inducing angiogenesis, by increasing endothelial nitric oxide (NO) [58]. BACE1 cleavage of DCC may, therefore, have important, and not yet understood, functions, which could be targeted through BACE1 inhibition to improve cerebrovascular function. 

The Netrin-1/DCC complex co-immunoprecipitates with APLP1, which led to identification of APP as a functional Netrin-1 receptor [59]. Furthermore, administration of Netrin-1 was found to decrease Aβ levels [59]. This further supports a link between BACE1 and DCC. Our analysis identified DCC as a BACE1 substrate altered in AD, showing that increased BACE1 activity likely causes a reduction in the full length DCC protein in AD. As the DCC/Netrin-1 interaction has been shown to play a role in the adult brain in axon guidance and angiogenesis, this may contribute to the neuronal and vascular impairments observed in AD. As Netrin-1 has been reported to have a similar affinity for APP, as DCC, increased BACE1 activity may lead to increased availability of Netrin-1, and therefore increased binding to APP [59]. Netrin-1 binding to APP has been shown to induce APP mediated signalling and gene transcription and suppress Aβ production. An alternative possibility is that the cleaved fragment of DCC plays a role in Netrin-1/APP binding, and that the subsequent effects on APP signalling and Aβ rely on the interaction with DCC. The cleavage of DCC would therefore prevent suppression of Aβ. The Netrin-1/DCC/APP interactions should be further investigated to fully understand the effects of BACE1 cleavage of DCC. Future analysis should aim to validate these changes and interactions in brain tissue. A greater understanding of this process may unveil a novel mechanism by which AD could be therapeutically targeted, as well as unknown physiological functions of BACE1.

#### 3.2.3. Cell Adhesion

Cell adhesion terms were frequently identified when the protein lists were subject to functional analysis, including Contactin associated protein 2 (CNTNAP2), Cell Adhesion Molecule 1 (CADM1), and Neurexin 3 (NRXN3), identified as BACE1 substrates differentially expressed in AD. Cell adhesion molecules (CAMs) mediate cell interactions with both neighbouring cells and surroundings. CAMs are vital in numerous cell processes, including cell survival, activation, cell to cell communication, interactions with the extracellular matrix (ECM), and cell migration. CAMs have also been implicated in disease, including neurological diseases such as AD. Altered neurogenesis is observed in AD, and has been linked to changes in cell adhesion molecules including the established BACE1 substrates neural cell adhesion molecule 1 (NCAM1) and L1 cell adhesion molecule (L1CAM) [60]. Changes in the cleavage of NCAM1 [61] and abundance of L1CAM in cerebrospinal fluid [62] is observed in AD. 

This analysis identified novel BACE1 substrates which may play important roles in AD, including CNTNAP2. CNTNAP2 is a novel BACE1 substrate that has previously been associated with AD, where it is found downregulated in the hippocampus [63]. CNTNAP2 has also been identified in human CSF, evidence for it undergoing ectodomain shedding in the brain [64]. CADM1 and NRXN2 have been shown to be associated with γ-secretase, suggesting similar regulation to APP. CADM1 has previously been shown to be sequentially cleaved by ADAM-10 and γ-secretase [61]. ADAM-10 and BACE1 have previously been shown to interact and regulate the cleavage of substrates including CHL-1 [65]. It is, therefore, possible that a similar process occurs in the cleavage of CADM1, with regulation of its shedding mediated by both BACE1 and ADAM-10. The ectodomain shedding of CADM1, and the intracellular domain (ICD) product of γ-secretase cleavage, has been suggested to be involved in transcriptional regulation as well as a mechanism of protein regulation [66]. NRXN3 is a member of the neurexin family, proteins important for synaptic function. Interestingly, NRXN3 has been shown to be sequentially cleaved by ɑ-secretase and γ-secretase, with the cleaved fragments suggested to play a role in regulation of neurexins and signal transduction at synapses [67]. Furthermore, AD-associated mutation of γ-secretase was found to impair NRXN3 processing. If BACE1 also cleaves NRXN3, this would suggest similar regulation of the cell adhesion molecule to APP, and could potentially be a pivotal mechanism in AD. It is, therefore, arguable that BACE1 is involved in the development and progression of AD in roles outside the production of Aβ. This analysis identified novel cell adhesion molecules likely to be cleaved by BACE1, further investigation of which may shed light on the involvement of BACE1 in changes in cell adhesion molecules in AD. The identification of CAMs that may be BACE1-dependent or substrates presents important new avenues to explore mechanisms of AD development and treatment.

#### 3.2.4. Ephrins and Ephrin Receptors

Ephrin receptor tyrosine kinases bind membrane bound ephrin ligands, and elicit bi-directional signalling into both cells [68]. Together, Ephrin receptors and Ephrins mediate short distance cell to cell communication, and can lead to both cell repulsion and adhesion, as well as changes in cell morphology. They are important in mediating the axon growth cone in axon guidance, synaptic plasticity, and in vasculogenesis. Ephrin-signalling-related terms, and ephrin receptors EphA4 and EphA2, came up frequently in our analysis. EphA4 and EphA2 have previously been identified as putative BACE1 substrates [36]. Furthermore, BACE1 has previously been observed to cleave EphA4 in HEK293 cells, although there was no evidence for BACE1 cleavage of EphA4 in hippocampus homogenates from BACE1^+/+^ and BACE1^−/−^ mice [66]. Despite not yet being successfully identified in vivo as a BACE1 substrate, our analysis shows that EphA4 expression is altered in human AD brain samples. As EphA4 has been validated as an in vitro BACE1 substrate, although not yet fully validated, our identification of EphA4 as a potential BACE1 substrate may present a novel mechanism of BACE1 action in health and disease. 

#### 3.2.5. Axon Guidance

A recurring functional term that appeared throughout our analysis was axon guidance, and axonogenesis. It is already known that BACE1 is involved in axon guidance, through its cleavage of neural cell adhesion molecule close homolog of L1 (CHL1) [69]. Our analysis highlighted DCC/Netrin-1 signalling [70], Robo/Slit signalling [71], and EphA4/EphrinB3 signalling [72], all associated with left and right locomotor function. ROBO2 and ROBO1 have also previously been identified as putative substrates, although not validated [36]. Deletion of these molecules has been linked to abnormal axon wiring, and a ‘hopping’ phenotype [72]. The presence of three signalling pathways known to be important for locomotor function in our analysis, strongly suggests a central role for BACE1 in the regulation of axon guidance and axonogenesis, beyond that of the cleavage of CHL1.

## 4. Materials and Methods

### 4.1. Computational Identification of BACE1 Substrates

We explored the literature for proteomic datasets of proteins differentially expressed (DE) in response to changes in BACE1, through overexpression, knockout, knockdown, or inhibition. From our initial list of potential publicly available datasets, we selected six datasets based on the data availability, the species, and the presence of values for a fold change, and adjusted *p*-value (Figure 1) (Table 3) [14,33,34,35,36]. The datasets were filtered by fold change <−1.3 or >1.3, and adjusted *p*-value (*p* < 0.05), with the exception of where the published data had already been filtered, in which case the entire list was taken forward for analysis. Duplicates were removed, and the remaining list of 533 BACE1-dependent proteins was then compared against a list of bioinformatically predicted BACE1 substrates, giving a list of 120 BACE1 substrates (Figure 1) [37]. This list of 120 BACE1 substrates was taken forward for further analysis.

### 4.2. Identification of Proteins Altered in Alzheimer’s Disease

The single nucleus AD RNA sequencing data was already set at an adjusted *p* < 0.1 and a log2 fold change ≥0.1 or ≤ −0.1; therefore, the entire list was taken forward [38]. The list of BACE1 substrates DE in AD was compared against genes DE in epilepsy, stroke, traumatic brain injury (TBI), and experimental autoimmune encephalitis (EAE), to decipher which proteins changed AD specifically [50]. Genes were filtered by a log2 fold change of either <−1 or >1, due to the lower sensitivity of proteomics than transcriptomics, and adjusted *p*-value (<0.05), and timepoints pooled.

### 4.3. Analysis of Protein Lists

The Bioinformatics and Evolutionary Genomics database was used to compare multiple lists of genes or proteins and create Venn diagrams. All lists were input as gene symbols, converted from GenBank accession IDs and UniProt symbols using DAVID. Venns were redrawn using PowerPoint. The analysis was conducted using Microsoft Excel, Enrichr, STRING, and GraphPad Prism. Enrichr provides a single platform to access dozens of online analysis tools to compare lists with thousands of datasets in the public domain and identify functional enrichment. Lists of gene symbols were input into the home page of Enrichr. Enriched terms were filtered by adjusted *p* value and exported as a text file for further processing in STRING or Prism. STRING builds networks based on known protein–protein interactions. Gene symbol lists were input into the STRING homepage and ‘Homo sapien’ selected. 

### 4.4. In Silico Modelling of BACE1 Substrate Interactions

The bioinformatically predicted BACE1 cleavage recognition sequences for PTPRD and DCC [37] were used to computationally model a predicted interaction for BACE1-PTPRD and BACE1-DCC. Whilst there are numerous structures available for both proteins, neither of the regions predicted to be the putative cleavage regions have structural information available. 

The AlphaFold Protein Structure Database was used to obtain a model for PTPRD [69,73]. We identified a complete and well modelled version of PTRD with the per-residue confidence score for the cleavage region between 83.92 and 72.58. We used this model in conjunction with the structure of BACE1 (PDB ID 1W50- Apo Structure of BACE (Beta Secretase)) [74] and the High Ambiguity Driven protein–protein DOCKing (HADDOCK) protein–protein docking webserver [39,40] to predict the molecular basis for the interaction. HADDOCK clustered 132 structures in 13 clusters, which represents 66% of the water-refined models generated for the PTPRD-BACE1 protein–protein interaction. 

The AlphaFold Protein Structure Database did not contain a model with a high degree of per-residue confidence score for the cleavage region of DCC (<50) so a model was made using the I-TASSER (Iterative Threading ASSEmbly Refinement) structure prediction server [75]. The best model for the DCC cleavage region using I-TASSER had a C-score of −2.41 and an estimated TM-score = 0.43 ± 0.14. C-score is a confidence score for estimating the quality of predicted models by I-TASSER. It is calculated based on the significance of threading template alignments and the convergence parameters of the structure assembly simulations. C-score is typically in the range of −5 to 2, where a C-score of higher value signifies a model with a high confidence. A TM-score > 0.5 indicates a model of correct topology and a TM-score < 0.17 means a random similarity. Using this model in conjunction with the structure of BACE-1 and the HADDOCK protein–protein docking webserver, we predicted the molecular basis for the interaction. HADDOCK clustered 154 structures in 7 clusters, which represents 77% of the water-refined models generated for the DCC-BACE1 protein–protein interaction. 

### 4.5. Mice

Mice were given free access to food and water and maintained on a 12 h light/dark cycle. BACE1^−/−^ mice, C57BL/6J background, were obtained from the Jackson Laboratory (B6.129-Bace1tm1Pcw/J) and BACE1^−/−^ and wild type control mice were generated using a BACE1^−/+^ X BACE1^−/+^ breeding strategy. Both male and female mice at 8–10 weeks of age were used for experiments. 

### 4.6. Primary Endothelial Cell Isolation and Protein Harvest

Primary brain endothelial cells (BECs) were isolated from BACE1 knockout mice and littermate wildtype controls. The brains were minced and incubated in 0.2 mg/mL collagenase dispase dissolved in DMEM (1 h, 37 °C, 5% CO_2_). The samples were homogenised using a cannula attached to a stripette, strained in a 70 uL cell strainer, centrifuged (1000× *g*, 10 min at room temperature), and resuspended in 5 mL 20% BSA solution. The samples were centrifuged (1000× *g*, 20 min, room temperature) the myelin layer discarded and resuspended in 5 ml PBS before a final centrifuge (400× *g*, 5 min, room temperature). The pellet was resuspended in MV2 media (Promocell, Heidelberg, Germany; C-22022), supplemented with antibiotic antimycotic solution (Merck, Darmstadt, Germany; A5955-100ML), and cells incubated (37 degrees, 5% CO_2_) until confluent with the media changed every 2–3 days. We expect >95% of remaining cells to be BECs [76]. Cells were grown on 0.2% gelatine-coated plates. Once confluent, BECs were washed with ice cold PBS and lysed with a cell extraction buffer (Fisher Scientific, FNN0011) supplemented with protease inhibitor (Merck, Darmstadt, Germany). 

### 4.7. Cell Culture 

hCMEC/D3 cells were grown on 0.2% gelatine-coated flasks/plates, and grown in ECGM2 (Endothelial Cell Growth Medium 2, Promocell) supplemented with antibiotic antimycotic solution (Merck, Darmstadt, Germany, A5955-100ML) at 37 degrees, and 5% CO_2_. For transfection, cells were plated on to 6-well plates at 100,000 cells per well and grown for 3 days. Cells were then transfected with 1ug of plasmid DNA of either empty vector (pcDNA3.1) or BACE1 per well, using Lipofectamine™ 2000 reagent (Invitrogen, Waltham, MA, USA), according to the manufacturer’s instructions. For BACE1-inhibitor treatments, cells were plated onto 6-well plates at 100,000 cells per well and grown for 3 days, and treated with Merck-3 (250 nM, Merck, Darmstadt, Germany) for 24 h. At 24 h post-transfection or inhibitor treatment, the cells were washed with ice cold PBS and lysed with a cell extraction buffer (Fisher Scientific, FNN0011) supplemented with protease inhibitor (Merck), or pelleted for RNA extraction. 

### 4.8. Western Blotting

Protein lysates (15 μg) were diluted with 4× NuPAGE™ LDS Sample buffer (Thermofisher, Waltham, MA, USA; NP0007) and 10× NuPAGE™ Sample reducing agent (Thermofisher, Waltham, MA, USA; NP0009) and heat denatured. They were then subject to SDS-PAGE electrophoresis on NuPAGE™ 4–12% Bis-Tris gels (Thermofisher, Waltham, MA, USA; NP0335BOX) using NuPAGE™ MES SDS Running Buffer (20×) (Thermofisher, Waltham, MA, USA; NP0002), before being transferred to polyvinylidene fluoride membranes (Merck, Darmstadt, Germany, IPVH00010). The membranes were blocked and incubated with the appropriate primary antibody diluted in 5% BSA TBS/Tween overnight at 4 °C. Primary antibodies included anti-NCAM1 (R&D Systems, anti-hNCAM-1/CD56 AF2408, 1:1000), anti-PTPRD (St Johns Laboratory—Anti-PTPRD antibody (STJ191298), 1:1000), anti-DCC (Abcam, Cambridge, UK; anti-DCC [EPR23313-115] (ab273570) 1:1000), and anti-beta actin (Santa Cruz, CA, USA; Anti-beta actin, SC-47778, 1:2500). HRP-conjugated secondary antibodies (ECL α-mouse HRP, Merck, Darmstadt, Germany; NA931V, 1:5000 dilution), and ECL α-rabbit HRP (Merck, Darmstadt, Germany; NA934V, 1:5000 dilution) were used for detection. Membranes were developed using ECL Immobilon reagent and imaged using a Gbox imager with Genesis Software (version 4.3.10.0, VWR, Leicestershire, UK). ImageJ software was used for densitometric analysis of the bands. The band density of the proteins of interest were normalised to the respective beta actin band, as loading control. These data were then expressed as a fold change relative to the protein expression in the control samples. Graphs were plotted using GraphPad Prism Software (Version 8.4.3, San Diego, CA, USA).

### 4.9. Quantitative RT-PCR 

Gene expression was determined by qRT-PCR. Total RNA was extracted from frozen cell pellets using the Monarch^®^ Total RNA Miniprep Kit (New England BioLabs, Ipswich, MA, USA). A total of 1 μg of total RNA was used for cDNA synthesis using Lunascript RT Supermix Kit (New England Biolabs, Ipswich, MA, USA). For quantitative PCR, cDNA was amplified (QuantStudio™ 3 Real-Time PCR System, Thermofisher, Waltham, MA, USA), using PrimePCR SYBR Green Assay, Desalt 200R primers and probes; PTPRD (qHsaCED0045607) and DCC (qHsaCED0047904). Genes of interest were normalised to GAPDH (qHsaCED0038674) as a housekeeping gene and are expressed relative to gene expression in the control group. 

### 4.10. Statistical Tests 

Data were tested for normality using the Shapiro-Wilk normality test, and all passed this test (*p* > 0.05). Consequently, significance (*p* ≤ 0.05) was determined by unpaired Student’s *t* test. The qRT-PCR data was log transformed using Y = Log(Y) before analysis. 

## 5. Conclusions

Together, this analysis presents numerous novel BACE1 substrates and functions that could be pivotal in understanding and treating AD. Ultimately, this underlines the importance of fully understanding the physiological functions of BACE1, including for the development of successful BACE1 inhibitors for the treatment of AD. Further investigation into the presented findings could lead to important and exciting findings for our knowledge of BACE1.

## Figures and Tables

**Figure 1 ijms-23-04568-f001:**
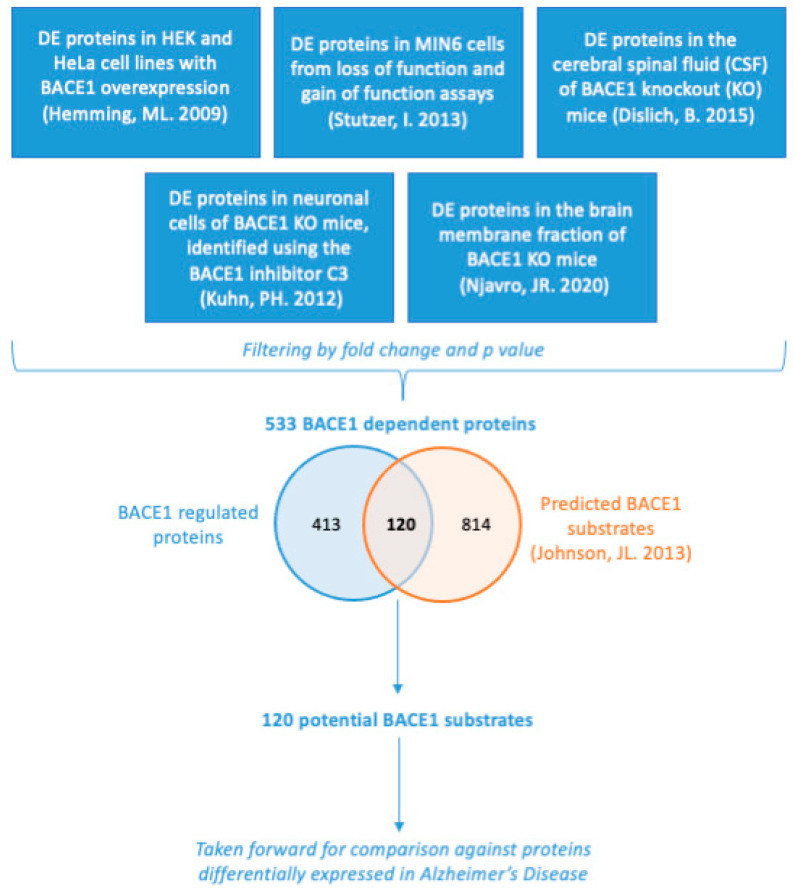
Summary of how the list of 533 BACE1-dependent proteins and 120 BACE1 potential substrates was obtained. Proteomics data from various studies acquired through datamining was filtered by fold change and *p* value and combined. This list of 533 BACE1-dependent proteins was then compared against a list of 934 bioinformatically predicted BACE1 substrates [37], to give a list of 120 BACE1 potential substrates which was the basis of further analysis [14,33,34,35,36].

**Figure 2 ijms-23-04568-f002:**
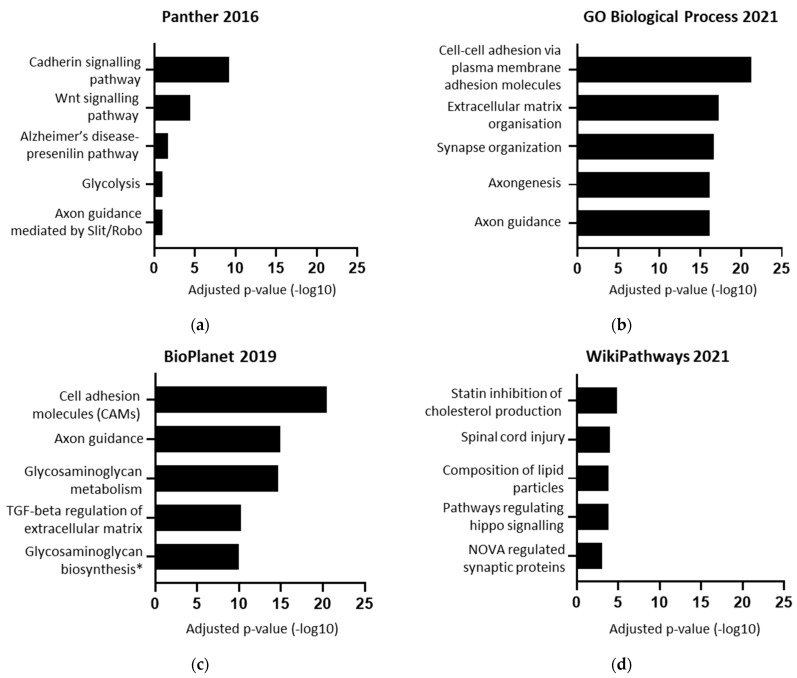
Functional ontology of 533 BACE1-regulated proteins. (**a**–**d**) Enrichr terms from various databases plotted by −log10 of the adjusted *p*-value. (**c**) * Specifically, tetrasaccharide linker sequence formation in glycosaminoglycan biosynthesis.

**Figure 3 ijms-23-04568-f003:**
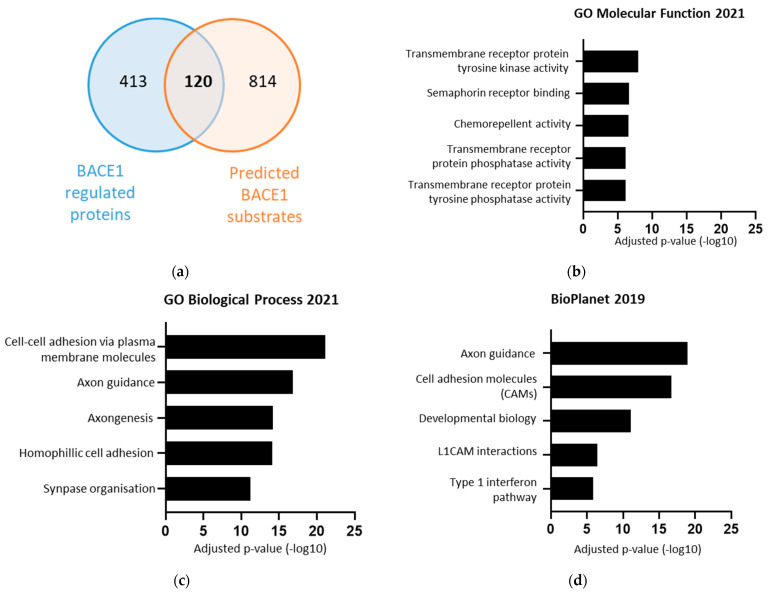
The 120 BACE1 substrates, identified from BACE1 experimental dependent proteins overlaid with BACE1 predicted substrates, were found to be associated with various functions including axon guidance and cell adhesion. (**a**) Venn diagram overlaying 533 experimental secreted BACE1 dependent proteins, and 934 bioinformatically predicted BACE1 substrates [37]. (**b**–**d**) Terms from various databases when the 120 BACE1 substrates where input into Enrichr, showing log of adjusted *p*-value.

**Figure 4 ijms-23-04568-f004:**
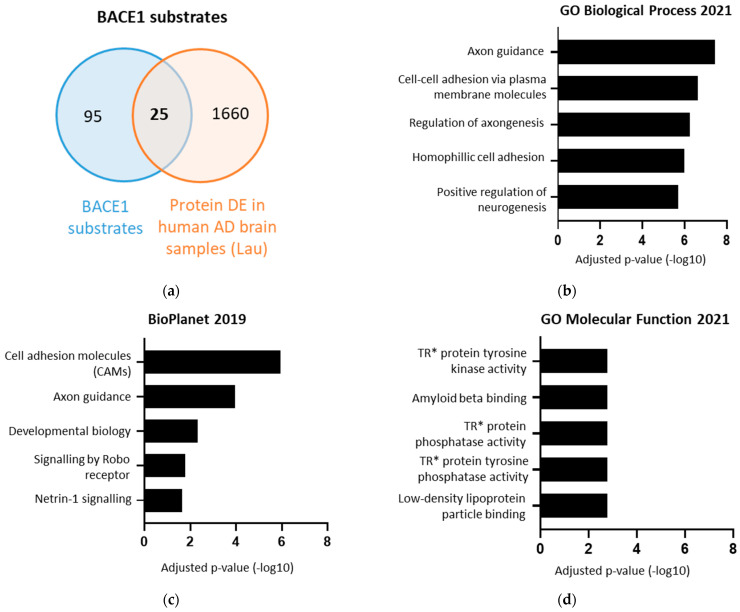
The 25 BACE1 substrates differentially expressed in Alzheimer’s disease (AD). (**a**) Venn diagram overlaying 120 BACE1 substrates with 1685 proteins differentially expressed in AD [38]. (**b**–**d**) Terms from various databases when the 25 BACE1 substrates DE in AD were input into Enrichr, showing log of adjusted *p*-value. * TR = transmembrane receptor.

**Figure 5 ijms-23-04568-f005:**
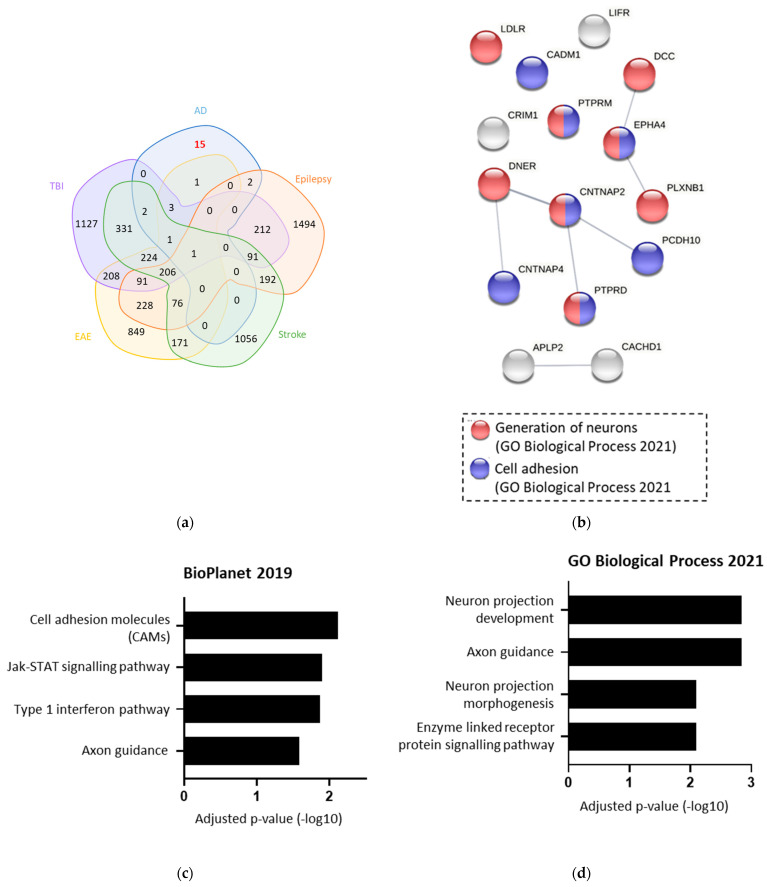
The 15 BACE1 substrates were Alzheimer’s disease (AD)-specific when compared against other diseases. (**a**) Venn diagram overlaying 25 BACE1 substrates differentially expressed (DE) in AD with proteins DE in traumatic brain injury (TBI), epilepsy, stroke, and experimental autoimmune encephalitis (EAE) (50)). (**b**,**c**) Terms from various databases when the 15 AD-specific BACE1 substrates were input into Enrichr, showing log of adjusted *p*-value. (**d**) String analysis of the 15 AD-specific BACE1 substrates, showing associations with GO terms.

**Figure 6 ijms-23-04568-f006:**
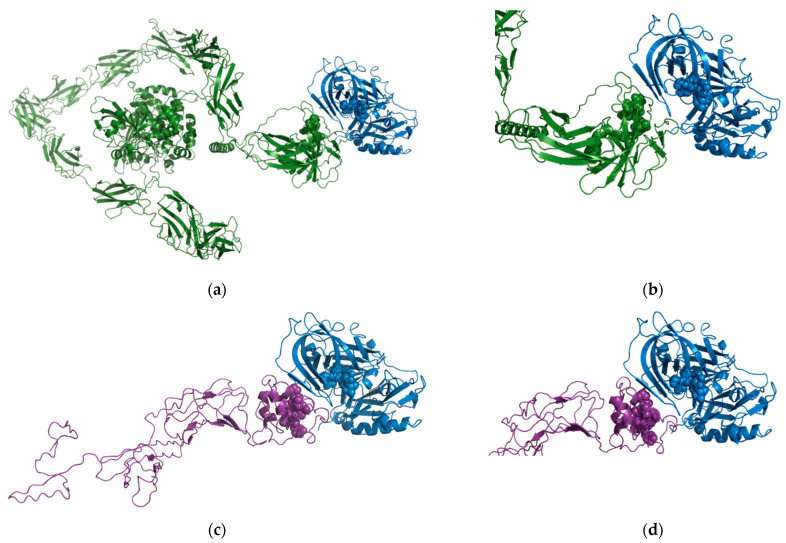
Predicted BACE1-PTPRD and BACE1-DCC interactions. (**a**) Overview of the predicted BACE1-PTPRD interaction. BACE1 (PDB ID 1W50) shown as blue ribbons with the active site highlighted as blue spheres and PTPRD (AlphaFold model AF-P23468-F1) shown as green ribbons with the predicted BACE1 cleavage region highlighted as green spheres. (**b**) Close-up of the predicted BACE1-PTPRD interaction. (**c**) Overview of the predicted BACE1-DCC interaction BACE1 (PDB ID 1W50) shown as blue ribbons with the active site highlighted as blue spheres and the domain of DCC predicted to be cleaved by BACE1 (modelled using I-TASSER) shown as purple ribbons with the predicted BACE1 cleavage region highlighted as purple spheres. (**d**) Close-up of the predicted BACE1-DCC interaction.

**Figure 7 ijms-23-04568-f007:**
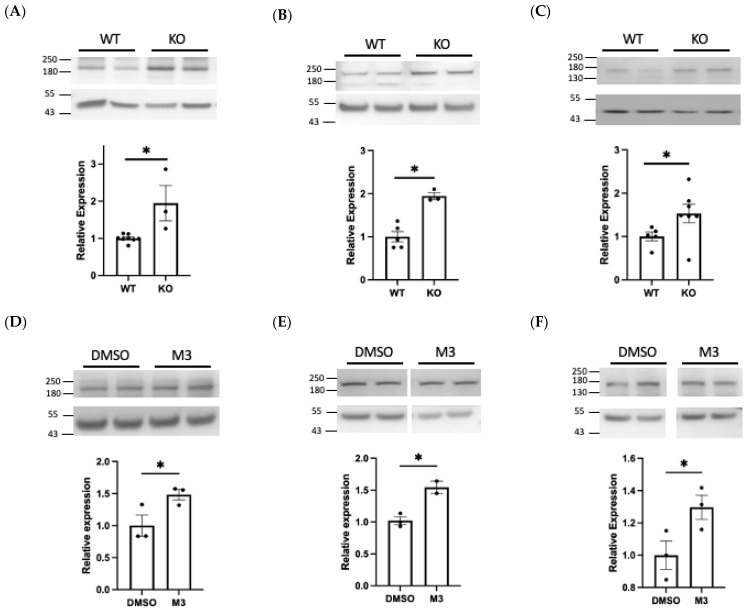
Experimental validation of NCAM1, PTPRD, and DCC as BACE1 substrates. Western blots were performed on protein lysates from BACE1 knockout (KO) and wildtype (WT) brain endothelial cells (BECs) or hCMEC/D3 treated with M-3 or transfected with BACE1. Relative protein expression of full length NCAM1 (**A**,**D**,**G**), PTPRD (**B**,**E**,**H**), and DCC (**C**,**F**,**I**) after normalisation to beta actin. PTPRD and DCC gene expression (**J**,**K**) in hCMEC/D3. N = 3 per group, * denotes *p* < 0.05.

**Table 1 ijms-23-04568-t001:** The 120 BACE1 substrate list. Identified from BACE1 experimental dependent proteins overlaid with BACE1 predicted substrate [37]. Validated and well characterised substrates are shown in red. The table is shaded grey where three or more members of the same family were found. Details can be searched on GeneCards: https://www.genecards.org/ accessed on 6 May 2020.

ACE	CNTNAP4	LINGO2	NLGN4X	PCDHGC3	SDK2
ACP2	CRIM1	LMAN2	NRCAM	PLXNB1	SEMA4A
ADAM22	CSF1R	LPL	NRP2	PLXNB2	SEMA4B
**ALCAM**	DCC	LRFN4	**NRXN3**	PODXL2	SEMA4C
ALG2	DNER	LRIG1	NTM	PTPRD	SEMA6A
**APLP2**	DPEP2	LRIG2	NTRK2	PTPRK	SEMA7A
**APP**	DSG2	LRIG3	OPCML	PTPRM	**SEZ6**
ATP6AP1	EGFR	LRP11	PCDH1	PTPRN	**SEZ6L**
ATP6AP2	EMB	LRP4	PCDH10	PTPRS	SLITRK1
**CACHD1**	EPHA2	LRRN1	PCDH17	PTPRT	SORCS1
CADM1	**EPHA4**	LYVE1	PCDH19	PTPRU	SORCS3
CADM4	EPHA7	MBTPS1	PCDH20	PVR	SORL1
CDH8	GLG1	MET	PCDH7	QSOX2	SSR1
CEACAM1	GPC3	MMP17	PCDH8	RGMA	TLR9
**CHL1**	GPC4	MPZL1	PCDH9	RGMB	TMEM132A
CLSTN2	IL6ST	**NCAM1**	PCDHGA11	ROBO1	TMEM132B
CLSTN3	**L1CAM**	NCSTN	PCDHGA12	ROBO2	TMEM132E
CNST	LAMP1	NEO1	PCDHGA4	RTN4RL1	TMX3
CNTNAP1	LDLR	NFASC	PCDHGA5	SDC4	VCAM1
CNTNAP2	LIFR	NLGN1	PCDHGA8	SDK1	VLDLR

**Table 2 ijms-23-04568-t002:** The 25 BACE1 substrates differentially expressed in Alzheimer’s disease (AD). Identified from BACE1 experimental dependent proteins overlaid with BACE1 predicted substrate [37], subsequently compared against proteins differentially expressed in brain samples from individuals with AD and other non-AD diseases (epilepsy, stroke, traumatic brain injury, experimental autoimmune encephalitis). BACE1 substrates that are AD-specific are shaded grey. Validated and well-characterised substrates are shown in red. Details can be searched on GeneCards: https://www.genecards.org/ accessed on 6 May 2020.

Protein Name	Gene Symbol
Activated Leukocyte Cell Adhesion Molecule	ALCAM
Amyloid Beta Precursor Like Protein 2	APLP2
VWFA and cache domain-containing protein 1	CACHD1
Cell Adhesion Molecule 1	CADM1
Contactin Associated Protein 2	CNTNAP2
Contactin Associated Protein Family Member 4	CNTNAP4
Cysteine Rich Transmembrane BMP Regulator 1	CRIM1
Netrin receptor DCC	DCC
Delta and Notch-like epidermal growth factor-related receptor	DNER
Ephrin type-A receptor 4	EPHA4
Low-density lipoprotein receptor	LDLR
LIF Receptor Subunit Alpha	LIFR
Lymphatic Vessel Endothelial Hyaluronan Receptor 1	LYVE1
Neurexin-3	NRXN3
Neurotrimin	NTM
Neurotrophic Receptor Tyrosine Kinase 2	NTRK2
Protocadherin-10	PCDH10
Plexin B1	PLXNB1
Protein Tyrosine Phosphatase Receptor Type D	PTPRD
Protein Tyrosine Phosphatase Receptor Type M	PTPRM
Roundabout Guidance Receptor 1	ROBO1
Roundabout Guidance Receptor 2	ROBO2
Syndecan 4	SDC4
Sidekick Cell Adhesion Molecule 1	SDK1
Sortilin Related Receptor 1	SORL1

**Table 3 ijms-23-04568-t003:** Publicly available datasets used for computationally identifying BACE1 substrate lists.

Reference	Study Type	Experimental Conditions	How Data Was Used	Statistical Analysis by Authors
[11]	SILAC proteomics	Brain membrane fraction from WT and BACE1 KO mice	Identify BACE1-regulated proteins	Log2 ratios of technical replicates were averaged and average protein log2 fold changes were calculated between BACE1 KO and WT samples. A two-sided Student’s t test evaluated the significance of proteins. Permutation-based FDR estimation was used.
[30]	SPECS proteomics	Primary neurons from BACE1 inhibitor treated, WT, and BACE1 KO mice	Identify BACE1-regulated proteins	A variance score (VS = absolute value of (standard error of the mean/(1 − mean))) was calculated for all proteins. Proteins with a vs. of ≤0.35 were considered as proteins with a consistent change upon BACE1 inhibition.
[31]	Loss/Gain-of-Function assays	MIN6 with knockdown of BACE1 and BACE2	Identify BACE1-regulated proteins	Protein significance analysis was performed using SRMstats where a constant normalisation was performed for all runs to equalise the median peak intensities of the heavy transitions from all the peptides between runs.
[32]	Quantitative proteomics	CSF from WT and BACE1 −/− mice	Identify BACE1-regulated proteins	Using the mean, the average LFQ intensity within each biological group was calculated. A two-sided t test was used, and the *p* value was corrected using false discovery rate (FDR)-based multiple hypothesis testing.
[33]	SILAC proteomics	HeLa/HEK with BACE1 overexpression	Identify BACE1-regulated proteins	Proteins containing peptides with at least 65% of the total signal derived from the BACE1 condition were considered as putative substrates. This threshold value is equivalent to a 1.857-fold increase in peptide abundance.
[34]	BACE1 substrate prediction	Bioinformatic analysis	Identify potential BACE1 substrates	
[35]	Single nucleus RNA sequencing	Single nucleus prefrontal cortical samples of AD patients and normal control (NC) subjects	Used to identify genes differentially expressed in Alzheimer’s disease	Data were background corrected and quantile normalised. Differential expression was performed via limma using a block design to leverage technical replicates. Genes with a false discovery rate (FDR)-adjusted *p* < 0.05 were considered differentially expressed. An algorithm based on the hypergeometric distribution is used to calculate enrichment *p* values.
[36]	RNA sequencing	Endothelial and brain cells from mouse models of stroke, multiple sclerosis (EAE), traumatic brain injury (TBI), and epilepsy	Identify genes differentially expressed in endothelial cells, dependent on disease	Stratified samples according to classification of AD and NC samples and compared the transcriptome profiles of individual cell types between AD and NC samples by the Wilcoxon rank-sum test using the *FindMarkers* function with the parameters *logfc.threshold = 0* and *test.use = wilcox*. The level of statistical significance for cell-type-specific transcriptomic changes was set at an adjusted *p* < 0.1 and a log2 fold change ≥0.1 or ≤−0.1.

## Data Availability

The data presented in this study are available on request from the corresponding author.

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
