# Peer review of "PTPRD and DCC Are Novel BACE1 Substrates Differentially Expressed in Alzheimer’s Disease: A Data Mining and Bioinformatics Study"

_ijms, 2022, doi:10.3390/ijms23094568_

Round 1

Reviewer 1 Report

In the paper entitled “PTPRD and DCC are novel BACE1 substrates differentially ex-2 pressed in Alzheimer’s disease: A data mining and bioinfor-3 matics study”, Taylor et al. used bioinformatics tools to search for new substrates for BACE1, an enzyme that is critically involved in Alzheimer's Disease. Moreover, the authors used suitable experimental approaches to validate some of the results obtained in silico. Overall, I find the manuscript to be well-written: the introduction provides sufficient knowledge in the subject's domain, the materials and methods are described in detail, the results are clearly presented and discussed. Please find below some minor issues:

Minor issues:

  1. Please double-check if the information in lines 35-38 is properly cited. It seems that the source is missing.
  2. Please check the spelling and the way of expression throughout the manuscript. Minor mistakes were observed. For example: line 63-64: “small quantity” in that context should be replaced with “small number”, “characterized” instead of “characterised”, “…which have important functions in the brain, which have been reviewed elsewhere” avoid repetition etc.
  3. Lines 426-427: in vivo and in vitro should be italicized

Author Response

Reviewer 1

Comments and Suggestions for Authors

In the paper entitled “PTPRD and DCC are novel BACE1 substrates differentially expressed in Alzheimer’s disease: A data mining and bioinformatics study”, Taylor et al. used bioinformatics tools to search for new substrates for BACE1, an enzyme that is critically involved in Alzheimer's Disease. Moreover, the authors used suitable experimental approaches to validate some of the results obtained in silico. Overall, I find the manuscript to be well-written: the introduction provides sufficient knowledge in the subject's domain, the materials and methods are described in detail, the results are clearly presented and discussed. Please find below some minor issues:

We thank the reviewer for their very positive comments. We have amended the manuscript in response the minor issues raised by this reviewer.

Minor issues:

  • Please double-check if the information in lines 35-38 is properly cited. It seems that the source is missing.

We have now added the original citations for the three mutations described.

  • Please check the spelling and the way of expression throughout the manuscript. Minor mistakes were observed. For example: line 63-64: “small quantity” in that context should be replaced with “small number”, “characterized” instead of “characterised”, “…which have important functions in the brain, which have been reviewed elsewhere” avoid repetition etc.

We have changed the comments highlighted, however, have left the English spelling of characterised.

  • Lines 426-427: in vivo and in vitro should be italicized

We have changed these to be italicized

Reviewer 2 Report

This manuscript describes the work of Taylor et al. regarding the identification of two novel BACE1 substrates within the framework of AD. The manuscript is clear and well written, and the work is well described, except for some minor specific issues (listed below).
I find two single points that I could use some clarification.
The first one regards significance and fold change criteria: - computational identification of BACE1 substrates (section 4.1) was performed with a p < 0.05 criteria -identification of altered proteins from single nucleus RNAseq (4.2) with p < 0.1 (a huge value on its own) - in the same section, the BACE1 substrates list was filtered at p < 0.05. Regarding fold change, the authors go from FC = +-1.3 to log2 FC=+-0.1 to log2 FC = +-1. No reason is provided for these apparently varying criteria, and this should be included.
The second, and potentially more worrisome, regards the major conclusion of the work. The authors find that NCAM1, PTPRD, and DCC, are upregulated in cells from BACE1 KO mice v. cells from wt animals. This is interpreted as coming from decreased cleavage of these proteins by BACE. - why was NCAM1 also included? is it a control? why this specific one? - were the FC determined from the WBs? how? - is proteolysis ablation the only way these proteins can be increased? I'm wondering if there are any genetic-level regulatory mechanisms that could justify their over-expression concomitantly with BACE gene removal - concluding that PTPRD and DCC are BACE substrates without showing that they are in fact substrates is a bit far-fetched. some experimental results should be included - are the proteins available? are their structures known well enough to at least verify (computationaly? that the substrates fit within BACE1's catalytic centre? A number of experiments/simulations are possible!
In keeping with this second point, I'm afraid I must recommend major revision to give the authors an opportunity to clarify this issue.

Specific issues Line 32, "neurotoxic Abeta" - Is Abeta straitghtforward neurotoxic? There is some literature regarding the possible biological role of this peptide. some physiological roles have been proposed and verified to different levels.
Lines 152-154, "list of proteins differentially expressed in the brains of AD patients post-mortem, identified using single nucleus transcriptomics (Figure 4A)" - i'm unfamiliar with single cell nucleus transcriptomics, but I guess it's a RNAseq technique. Do all identified transcripts give rise to functional proteins? aren't there any known regulatory mechanisms for these 25 transcripts?
Line 451, and in many occurrences throughout the text, one can read "adjusted p-value" - what is this adjustment? at least 3 different possible "adjustments" come to mind that would fit nicely in this work.
Line 481, "Primary brain endothelial cells (BECs) were isolated from BACE1 knockout mice and  littermate wildtype controls." - information about the mice used in this study would be welcome - age, sex, etc. Also, wheat's the source of the knockout animals? Are they commercial or in-house?

Author Response

We thank the reviewer for their very useful and insightful comments and suggestions designed to improve our manuscript, which we have endeavoured to address. Our apologies for the delay in returning this manuscript, which was primarily caused by the derivation of new data sets (additional in vitro work and molecular simulations).

This manuscript describes the work of Taylor et al. regarding the identification of two novel BACE1 substrates within the framework of AD. The manuscript is clear and well written, and the work is well described, except for some minor specific issues (listed below).

I find two single points that I could use some clarification.

The first one regards significance and fold change criteria:

  • computational identification of BACE1 substrates (section 4.1) was performed with a p < 0.05 criteria

This was our own filtering of the proteomic lists (see below)

  • identification of altered proteins from single nucleus RNAseq (4.2) with p < 0.1 (a huge value on its own)

This was an already processed dataset (see below)

  • in the same section, the BACE1 substrates list was filtered at p < 0.05. Regarding fold change, the authors go from FC = +-1.3 to log2 FC=+-0.1 to log2 FC = +-1. No reason is provided for these apparently varying criteria, and this should be included.

The transcriptomic and proteomic data was filtered with a different fold change value due to the difference in the sensitivity. Transcriptomic data was filtered by fold change (FC) = +/-1 and proteomic data by FC = +/-1.3, to account for the lower sensitivity. Both were filtered by p < 0.05. We have added text to clarify which datasets were filtered via which criteria, and why.

Where datasets were provided in an already filtered format, we took the whole list forward based upon the author’s criteria. This was because we felt that the lists were sufficiently reduced, and that the authors justification for these criteria were valid. The purpose of applying these criteria to all datasets was simply to reduce the data into (1) proteins more likely to be truly changed and (2) to reduce the lists of a more manageable size, in order to identify proteins of interest. The purpose of the exercise was therefore not to specifically identify proteins of interest, but to reduce the size of the datasets into lists of proteins more likely to be significantly altered to a physiologically important extent.

On line 514 we have made a reference to the variance in the log2 FC cut off for the proteomics data in comparison to transcriptomic data.

On lines 491 - 494 we have clarified where the data had already been filtered, we did not filter it further.

 The second, and potentially more worrisome, regards the major conclusion of the work. The authors find that NCAM1, PTPRD, and DCC, are upregulated in cells from BACE1 KO mice v. cells from wt animals. This is interpreted as coming from decreased cleavage of these proteins by BACE.

We understand the reservations of the reviewer, however we have no performed additional in vitro and in silico experiments to support our hypothesis (described below) which we hope addresses the concerns.

  • why was NCAM1 also included? is it a control? why this specific one?

NCAM1 was identified as a potential BACE1 substrate at the beginning of our computational analysis, and was subsequently validated by another group in 2021 (Kim et. al. J Biol Chem. 2021). We therefore used it as a positive control for both the computational work (showing it identified true BACE1 substrates), and for changes in the endothelial cells.

  • were the FC determined from the WBs? how?

Protein expression was calculated via densitometry of the relevant protein bands analysed using Image J software. These values were normalised against the respective beta actin band to account for loading. The data were then expressed as a fold change relative to the protein expression in the control samples (either wildtype littermates, vehicle treatment or EV transfection). To clarify this, I have added a sentence in the methods (lines 611-614).

  • is proteolysis ablation the only way these proteins can be increased? I'm wondering if there are any genetic-level regulatory mechanisms that could justify their over-expression concomitantly with BACE gene removal

We agree additional regulatory mechanisms may result in the increase in the predicted BACE1 substrates, including increase gene expression. To address this we have performed additional experiments to measure gene expression of PTPRD and DCC following BACE1 inhibitor treatment or BACE1 overexpression. We find no change in gene expression despite changes in protein expression of the full length proteins. This suggests the increase in PTPRD and DCC is via a post-translational mechanism.

  • concluding that PTPRD and DCC are BACE substrates without showing that they are in fact substrates is a bit far-fetched. some experimental results should be included

We understand further experimental data would strengthen our conclusions that PTPRD and DCC are BACE1 substrates. As such we have undertook in vitro and in silico experiments to support this.

Firstly the bioinformatics analysis by Johnson et al, used in our article, predicted cleavage recognition sequences and cleavage recognition sites for PTPRD and DCC based upon known substrates of BACE1. Using this information, we have modelled the interaction of these regions with the BACE1 active site to determine whether an interaction would be possible. This approach confirmed PTPRD and DCC could interact with BACE1 and supports the idea that they could be BACE1 substrates

In addition to full length PTPRD and DCC protein levels increasing in BACE1 -/- endothelial cells we have generated additional data demonstrating a similar increase in endothelial cells treated with a BACE1 inhibitor. This treatment was 24 hours but starts to address any compensatory mechanisms that may cause the increases in expression owing to lacking BACE1 constitutively.

In addition we have transfected hCMEC/D3 to overexpress BACE1. We found a decreased level of full length PTPRD, DCC and NCAM1 protein. This overexpression data is suggestive of an increased cleavage of the proteins by BACE1.

Taken together this additional data, including the gene expression data, we believe shows that PTPRD and DCC are likely to be true BACE1 substrates.

We have added additional information on the prediction cleavage site of DCC and PTPRD into the discussion (lines 330-332). The in silico modelling has been included as new Figure 6 and the additional in vitro data has been included in Figure 7.

  • are the proteins available? are their structures known well enough to at least verify (computationaly? that the substrates fit within BACE1's catalytic centre? A number of experiments/simulations are possible!

The whole structures of DCC and PTPRD aren’t available. However based on the parts of the structure that is known, we have predicted their full structure and modelled in silico this with the BACE1 catalytic centre. As mentioned above we performed in silico modelling and generated  HADDOCK scores which were indicative that the identified cleavage sequences in PTPRD and DCC could interact with the active site of BACE1, and had similar scores to that of the well characterised substrate, APP. This modelling is included in Figure 6 and explanations are in section 2.4

In keeping with this second point, I'm afraid I must recommend major revision to give the authors an opportunity to clarify this issue.

We hope the additional explanation and experiments have clarified any issues.

Specific issues

Line 32, "neurotoxic Abeta" - Is Abeta straitghtforward neurotoxic? There is some literature regarding the possible biological role of this peptide. some physiological roles have been proposed and verified to different levels.

We appreciate Abeta has physiological roles so have removed the use of ‘neurotoxic’ in line 32, but kept it in in line 33 so that it is only used in reference to Abeta in the specific context (accumulation of Abeta into plaques in Alzheimer’s Disease). 

Lines 152-154, "list of proteins differentially expressed in the brains of AD patients post-mortem, identified using single nucleus transcriptomics (Figure 4A)" - i'm unfamiliar with single cell nucleus transcriptomics, but I guess it's a RNAseq technique. Do all identified transcripts give rise to functional proteins? aren't there any known regulatory mechanisms for these 25 transcripts?

Single nucleus transcriptomics is an RNAseq technique used to profile gene expression in cells which are hard to isolate, therefore similar to single cell RNAseq.

The 25 transcripts we identified are all protein coding and give rise to the 25 proteins described in Table 2 (referenced on line 160). We appreciate that there are additional regulatory mechanisms for some of these proteins, however we are proposing that BACE1 may be a novel method of regulation of these proteins.

Line 451, and in many occurrences throughout the text, one can read "adjusted p-value" - what is this adjustment? at least 3 different possible "adjustments" come to mind that would fit nicely in this work.

We have added a column to Table 3 (information on the datasets used) describing the statistical analysis that the authors have done, including how they calculated the adjusted p-value.

Line 481, "Primary brain endothelial cells (BECs) were isolated from BACE1 knockout mice and  littermate wildtype controls." - information about the mice used in this study would be welcome - age, sex, etc. Also, wheat's the source of the knockout animals? Are they commercial or in-house?

We have added clarification regarding the BACE1 -/- mice into the methods section (section 4.4)

Round 2

Reviewer 2 Report

The authors have put a nice effort in dressing all previous comments, including some new work, which I feel strongly improves this work. I would like to thank them for such effort.

I believe the manuscript is ready for publication, in its present form.